# QA-LoRA: Quantization-Aware Low-Rank Adaptation of Large Language Models

**Yuhui Xu**   **Lingxi Xie** ✉   **Xiaotao Gu**   **Xin Chen**   **Heng Chang**
**Hengheng Zhang**   **Zhensu Chen**   **Xiaopeng Zhang**   **Qi Tian**
Huawei Inc.          (✉: corresponding author)
{xyh6666,198808xc,guxt1994,chenxin061,changh.heng}@gmail.com
{imhmhm,chenzhengsu1,zxphistory}@gmail.com,    tian.qi1@huawei.com

## Abstract

Recently years have witnessed a rapid development of large language models (LLMs). Despite the strong ability in many language-understanding tasks, the heavy computational burden largely restricts the application of LLMs especially when one needs to deploy them onto edge devices. In this paper, we propose a quantization-aware low-rank adaptation (**QA-LoRA**) algorithm. The motivation lies in the imbalanced numbers of parameters for quantization and adaptation, and the solution is to use group-wise operators to increase the number of parameters for quantization meanwhile decreasing that of adaptation. QA-LoRA is easily implemented with a few lines of code, and it equips the original LoRA with two-fold abilities: (i) during fine-tuning, the LLM's weights are quantized (*e.g.*, into INT4) to reduce time and memory usage; (ii) after fine-tuning, the LLM and auxiliary weights are naturally integrated into a quantized model without loss of accuracy. We apply QA-LoRA to the LLaMA and LLaMA2 model families and validate its effectiveness in different fine-tuning datasets and downstream scenarios. The code is made available at https://github.com/yuhuixu1993/qa-lora.

## 1 Introduction

Recently, large language models (LLMs) (Brown et al., 2020; Scao et al., 2022; Zhang et al., 2022; Touvron et al., 2023a; Chowdhery et al., 2022; OpenAI, 2023; Zeng et al., 2023) have shown unprecedented performance across a wide range of language understanding tasks (Wei et al., 2022a) and served as the foundation of state-of-the-art chat systems (Bubeck et al., 2023). The diversity of real-world applications calls for a pipeline in which LLMs can be fine-tuned to fit different scenarios and quantized to be deployed onto edge devices (*e.g.*, mobile phones), and the key issue is to get rid of the heavy computational burden brought by the large number of parameters of LLMs.

There are two lines of research for this purpose. **The first one** is parameter-efficient fine-tuning (PEFT) (Houlsby et al., 2019; Li & Liang, 2021; Liu et al., 2021; He et al., 2022; Hu et al., 2021) which introduced a small number of learnable parameters while keeping most pre-trained parameters unchanged. Among them, low-rank adaptation (LoRA) (Hu et al., 2021), a popular PEFT algorithm, proposed to fine-tune low-rank matrices to complement the pre-trained weights. Despite the comparable performance to full-parameter fine-tuning, the memory usage of LoRA is still large, especially when the base LLM is large (*e.g.*, LLaMA-65B). **The second one** studies parameter quantization (Yao et al., 2022; Dettmers et al., 2022; Wei et al., 2022b; Frantar et al., 2023; Lin et al., 2023; Xiao et al., 2023; Dettmers et al., 2023b) where the trained weights are quantized into low-bit integers or floating point numbers. Although these methods can alleviate the computational burden, they often report unsatisfying accuracy especially when the quantization bit width is low.

Hence, it is an important topic to integrate PEFT with quantization. A naive solution is to perform post-training quantization (PTQ) after PEFT, but it reports unsatisfying accuracy especially when the quantization bit width is low. Advanced methods exist, but they are either computationally expensive in the fine-tuning stage (Liu et al., 2023) or unable to maintain the quantized property after fine-tuning (Dettmers et al., 2023a). In this paper, we propose a simple yet effective method for quantization-aware low-rank adaptation (**QA-LoRA**). Our idea is based on the imbalanced numbers

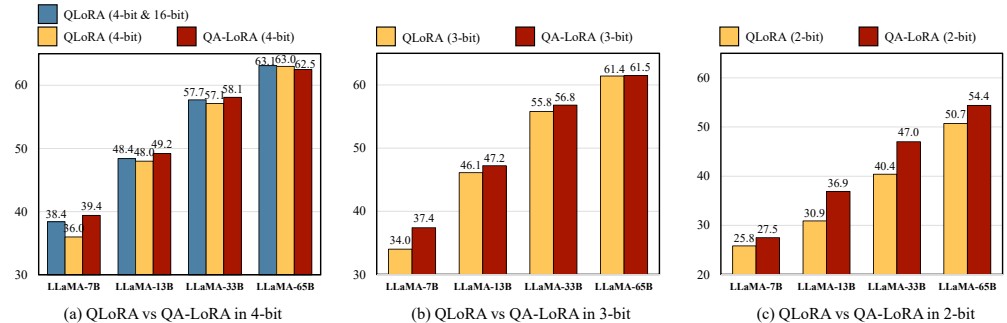

Figure 1: The comparison of 5-shot MMLU accuracy (%) with different quantization bit widths based on the LLaMA model family. QLoRA (NF4 & FP16) refers to the original QLoRA models with pre-trained weights in INT4 and adapter weights in FP16, and QLoRA (INT4) refers to performing post-training quantization (into INT4) upon the merged QLoRA models. All models are fine-tuned on the Alpaca dataset. Full results are provided in Table 1.

of parameters for quantization and adaptation. Specifically, each column of the pre-trained weight matrix is accompanied by only one pair of scaling and zero parameters but many more LoRA parameters. This imbalance not only results in large quantization errors (which harm the LLM's accuracy), but also makes it difficult to integrate the auxiliary weights into the main model. QA-LoRA addresses the issue by introducing group-wise operators to increase the number of parameters for low-bit quantization (each group is quantized individually) and decrease that of LoRA (each group shares the adaptation parameters). QA-LoRA enjoys two-fold benefits: (i) an efficient fine-tuning stage thanks to the LLM's weights being quantized into low-bit integers; (ii) a lightweight, fine-tuned model without the need for PTQ which often incurs loss of accuracy.

We evaluate QA-LoRA on the LLaMA and LLAMA2 model families (Touvron et al., 2023a;b) and validate it on various language understanding benchmarks. Figure 1 shows the comparison of 5-shot accuracy on the MMLU benchmark between QA-LoRA and the direct baseline, QLoRA (Dettmers et al., 2023a). QA-LoRA consistently outperforms QLoRA with PTQ on top of LLMs of different scales (the advantage becomes more significant when the quantization bit width is lower) and is on par with QLoRA without PTQ. Note that during inference, QA-LoRA has exactly the same complexity as QLoRA with PTQ and is much more efficient than QLoRA without PTQ. Hence, QA-LoRA is an effective and off-the-shelf method for joint quantization and adaptation of LLMs.

## 2 RELATED WORK

**Large language models (LLMs)** (Devlin et al., 2019; Brown et al., 2020; Zhao et al., 2023a; Hadi et al., 2023; Yu et al., 2023) have emerged as a dominant paradigm in natural language processing. It achieved the state-of-the-art on various tasks (Zhao et al., 2023b; Zhou et al., 2023; Wang et al., 2023) and served as the fundamental of chat systems (OpenAI, 2023). However, their deployment in real-world scenarios is hindered by the high computational and memory requirements during inference (Chang et al., 2023). To tackle this issue, various methods have been proposed, including distillation (Liu et al., 2023), quantization (Yao et al., 2022; Dettmers et al., 2022; Wei et al., 2022b; Frantar et al., 2023; Lin et al., 2023; Xiao et al., 2023), pruning (Frantar & Alistarh, 2023; Ma et al., 2023; Sun et al., 2023), *etc.* (Weng, 2023). This paper mainly focuses on the quantization of LLMs.

**Fine-tuning LLMs with adapters.** Parameter efficient fine-tuning (PEFT) is an important topic for LLMs. One of the most popular approaches is low-rank adaptation (LoRA) (Hu et al., 2021; Valipour et al., 2022), where the key insight is to decompose the adapter weights into the multiplication of two low-rank (and thus parameter-efficient) matrices. LoRA has claimed comparable performance to full fine-tuning while using much fewer learnable parameters. Meanwhile, there are also other branches of adapters for LLMs such as the series adapter (Houlsby et al., 2019) and parallel adapter (He et al., 2022). Please refer to (Mangrulkar et al., 2022; Hu et al., 2023) for a review of these adapters.

**Quantization of LLMs.** Quantization is a compression technique that reduces the bit width of the parameters and/or activations of LLMs to improve their efficiency and scalability (Xiao et al., 2023; Dettmers et al., 2022; 2023a). Existing methods mostly focused on preserving or restoring the accuracy of quantized LLMs during the inference stage (Zhu et al., 2023), where the key is to

reduce the memory footprint and computational costs without re-training the LLMs. One of the main challenges is to handle the outliers in the parameter distribution (Xiao et al., 2023), which can cause significant errors when quantized. To address this issue, some methods proposed to use either adaptive or dynamic quantization schemes that adjust the quantization range or precision according to the parameters (Xiao et al., 2023; Dettmers et al., 2022). Other methods used sophisticated grouping or clustering techniques to partition the parameters into different groups and applied different quantization strategies for each group (Park et al., 2022; Yao et al., 2022; Wu et al., 2023).

**Joint adaptation and quantization.** This paper aims to achieve the objectives of both parameter-efficient adaptation and computation-efficient tuning and deployment, which can further improve the efficiency and scalability of LLMs as well as mitigate the negative impact of quantization errors. However, this also poses additional challenges, such as propagating gradients through discrete values and optimizing the quantization parameters. To overcome these challenges, lossy quantization methods proposed to use stochastic rounding (Shen et al., 2020) or learned rounding (Esser et al., 2019) to approximate the gradients and update the parameters, but applying these methods to LLMs is often difficult. Other methods proposed to use switchback layers (Wortsman et al., 2023) or mixed-precision inference (Dettmers et al., 2023a) to alternate between quantized and full/half-precision values, which often result in low inference speed.

To the best of our knowledge, the most related work is QLoRA (Dettmers et al., 2023a) which squeezed the pre-trained weights into NF4 and added LoRA. However, QLoRA added the adaption weights back to pre-trained weights and turned them into FP16 again, and thus the deployed model is still slow. We solve this problem with the proposed QA-LoRA approach.

## 3 THE PROPOSED APPROACH

### 3.1 BASELINE: LOW-RANK ADAPTATION AND LOW-BIT QUANTIZATION

We follow the notation system used in LoRA (Hu et al., 2021) which assumed pre-trained weights to form a matrix $\mathbf{W}$ and the features form a vector $\mathbf{x}$. The definition is easily applied to a wide range of scenarios and extended into $\mathbf{x}$ is a set of vectors (*e.g.*, a feature matrix). Let the size of $\mathbf{W}$ be $D_{in} \times D_{out}$ and $\mathbf{x}$ has the length of $D_{in}$, and thus the computation is easily written as $\mathbf{y} = \mathbf{W}^\top \mathbf{x}$ where $\mathbf{y}$ is the output vector with a length of $D_{out}$.

The key idea of LoRA is to introduce a pair of matrices, $\mathbf{A}$ and $\mathbf{B}$, to supplement $\mathbf{W}$. $\mathbf{A}$ and $\mathbf{B}$ have sizes of $D_{in} \times D_{int}$ and $D_{int} \times D_{out}$, respectively, so that their multiplication, $\mathbf{AB}$, has the same size as $\mathbf{W}$. The intermediate dimensionality is often small (*i.e.*, $D_{int} \ll \min\{D_{in}, D_{out}\}$), making $\mathbf{AB}$ a low-rank matrix compared to $\mathbf{W}$. During fine-tuning, we compute $\mathbf{y} = \mathbf{W}^\top \mathbf{x} + s \cdot (\mathbf{AB})^\top \mathbf{x}$, where $s$ is the coefficient for weight tuning, and $\mathbf{W}$ is fixed while $\mathbf{A}$ and $\mathbf{B}$ can be adjusted, arriving at the goal of parameter-efficient fine-tuning. After fine-tuning, the computation is reformulated into $\mathbf{y} = (\mathbf{W} + s \cdot \mathbf{AB})^\top \mathbf{x}$, where $\mathbf{W}$ is replaced by $\mathbf{W}' = \mathbf{W} + s \cdot \mathbf{AB}$ for fast inference.

Another effective way to reduce computational costs lies in low-bit quantization. We only consider the quantization of weights throughout this paper. In particular, we apply a simple method named min-max quantization. Mathematically, given the bit width $N$ and a pre-trained weight matrix $\mathbf{W}$, we compute the minimum and maximum values across all elements of $\mathbf{W}$, denoted as $\min(\mathbf{W})$ and $\max(\mathbf{W})$, respectively. Then, $\mathbf{W}$ is quantized into $\tilde{\mathbf{W}}$ by computing

$$\tilde{\mathbf{W}} = \alpha \cdot \hat{\mathbf{W}} + \beta \doteq \alpha \cdot \left\lfloor \frac{\mathbf{W} - \beta}{\alpha} \right\rceil + \beta, \tag{1}$$

where $\alpha = (\max(\mathbf{W}) - \min(\mathbf{W}))/(2^N - 1)$ and $\beta = \min(\mathbf{W})$ are called the scaling and zero factors, respectively; $\lfloor \cdot \rceil$ denotes the integer rounding operation. All elements in $\hat{\mathbf{W}}$ are in the set of $\{0, 1, \ldots, 2^N - 1\}$ and thus stored as $B$-bit integers. The computation, $\mathbf{y} = \mathbf{W}^\top \mathbf{x}$, is approximated as $\mathbf{y} = \tilde{\mathbf{W}}^\top \mathbf{x} = \alpha \cdot \left\lfloor \frac{\mathbf{W} - \beta}{\alpha} \right\rceil^\top \mathbf{x} + \beta \mathbf{x}$. The quantization brings two-fold benefits, namely, the storage of $\mathbf{W}$ is reduced (*e.g.*, from FP16 to INT4) and the computation of $\mathbf{W}^\top \mathbf{x}$ becomes faster. The cost is that $\tilde{\mathbf{W}}$ is an approximation of $\mathbf{W}$, which may harm the accuracy of language understanding.

To reduce the quantization loss between $\mathbf{W}$ and $\tilde{\mathbf{W}}$, an effective strategy is to perform an individual quantization for each column of $\mathbf{W}$. Let $\mathbf{W} = [w_{i,j}]_{D_{in} \times D_{out}}$, where $i \in \{1, \ldots, D_{in}\}$ and $j \in$

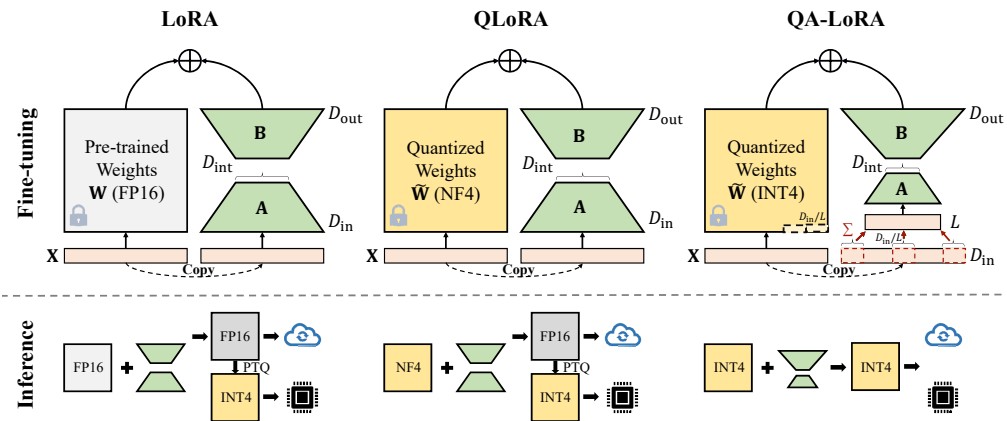

Figure 2: An illustration of the goal of QA-LoRA. Compared to prior adaptation methods, LoRA and QLoRA, our approach is computationally efficient in both the fine-tuning and inference stages. More importantly, it does not suffer an accuracy loss because post-training quantization is not required. We display INT4 quantization in the figure, but QA-LoRA is generalized to INT3 and INT2.

$\{1, \dots, D_{\text{out}}\}$ are iterative variables. Let $\alpha_j$ and $\beta_j$ be the scaling and zero factors computed on the $j$-th column, $\mathbf{w}_j$. Hence, Equation 1 is updated as $\tilde{\mathbf{W}} = [\tilde{\mathbf{w}}_j]_{D_{\text{out}}} = \left[\alpha_j \cdot \left\lfloor \frac{\mathbf{w}_j - \beta_j}{\alpha_j} \right\rceil + \beta_j \right]_{D_{\text{out}}}$,

and the computation is rewritten as $\mathbf{y} = \tilde{\mathbf{W}}^\top \mathbf{x} = \left[\alpha_j \cdot \left\lfloor \frac{\mathbf{w}_j - \beta_j}{\alpha_j} \right\rceil^\top \mathbf{x} + \beta_j \mathbf{x} \right]_{D_{\text{out}}}$. Compared to the original (holistic) quantization, the computational cost is unchanged while the storage cost of the scaling and zero factors increases from 2 to $2D_{\text{out}}$ floating point numbers. This is negligible compared to the reduced cost of storing the full-precision $\mathbf{W}$.

## 3.2 Objective: Efficient Adaptation and Deployment

As shown in Figure 2, we aim to achieve two goals. First, during the fine-tuning stage, the pre-trained weights $\mathbf{W}$ are quantized into low-bit representation so that LLMs can be fine-tuned on as few GPUs as possible. Second, after the fine-tuning stage, the fine-tuned and merged weights $\mathbf{W}'$ are still in a quantized form so that LLMs can be deployed with computational efficiency.

We note that QLoRA (Dettmers et al., 2023a), a recently proposed variant of LoRA, achieved the first goal. The idea is to quantize $\mathbf{W}$ from FP16 to NF4 (a highly squeezed type of floating point numbers) during the fine-tuning stage. We learn from QLoRA that joint optimization of quantization and adaptation is tractable because the accuracy loss between $\mathbf{W}$ and $\tilde{\mathbf{W}}$ is compensated by the low-rank weights, $s \cdot \mathbf{AB}$. After fine-tuning, the side weights $s \cdot \mathbf{AB}$ must be added back to $\tilde{\mathbf{W}}$, making the final weights $\mathbf{W}'$ in FP16 again. Indeed, one can perform post-training quantization (PTQ) upon $\mathbf{W}'$, but this strategy can cause a significant loss in accuracy especially when the bit width is low. Please refer to the experiments for details. Additionally, there is no operator-level optimization for NF4 yet, making it difficult to accelerate the fine-tuning and inference stages. *In brief, the only benefit brought by QLoRA is the reduced memory cost for fine-tuning.*

## 3.3 Solution: Group-wise Quantization with Low-rank Adaptation

From the above analysis, the key to achieving the second goal lies in that $\tilde{\mathbf{W}}$ (*i.e.*, the quantized $\mathbf{W}$) and $s \cdot \mathbf{AB}$ can be merged without using high-precision numbers (*e.g.*, FP16). We first note that this is impossible in the original setting, *i.e.*, $\mathbf{W}$ is quantized into $\tilde{\mathbf{W}}$ in a column-wise manner while both $\mathbf{A}$ and $\mathbf{B}$ are unconstrained.

We write down the condition using the mathematical language. Since $\mathbf{W}' = \tilde{\mathbf{W}} + s \cdot \mathbf{AB}$, we have $w'_{i,j} = \tilde{w}_{i,j} + s \cdot \sum_k a_{i,k} b_{k,j}$ for all $(i, j)$. Here, for any $j$, all $\tilde{w}_{i,j}$ are represented using the same set of scaling and zero factors, *i.e.*, there exist $\alpha_j$ and $\beta_j$ so that $\tilde{w}_{i,j} = \alpha_j \times \hat{w}_{i,j} + \beta_j$, $\hat{w}_{i,j} \in \{0, 1, \dots, 2^N - 1\}$. After each $\tilde{w}_{i,j}$ is added by $s \cdot \sum_k a_{i,k} b_{k,j}$ (abbreviated as $c_{i,j}$), if we want to keep the property for quantization, we must guarantee that for any $j$, all possible values of $c_{i,j}$

---

**Algorithm 1** QA-LoRA Pseudocode in the PyTorch-like style

```
# D_in, D_out, D_int: the input, output, and low-rank adaptation dimensions
# L: the quantization group numbers of weights W (D_in // L is the group size)
# s: the coefficient for adaptation; N: the bit width of quantization

QA = nn.AvgPool1d(D_in//L)
lora_A = nn.Parameter(torch.empty((D_int, L)))
lora_B = nn.Parameter(torch.empty((D_out, D_int)))

def qalora_forward(x, W, lora_A, lora_B):
    W_tilde = pre_quantization(W, alpha, beta)
    result = x @ W_tilde
    result += (QA(x)*(D_in//L)) @ lora_A.transpose(0,1) @ lora_B.transpose(0,1) * s
    return result

def pre_quantization(W, alpha, beta):
    W_hat = torch.round(W / alpha) + beta
    return alpha * (W_hat - beta)

def merge_with_quantization(beta, lora_A, lora_B):
    beta_new = beta - s * (lora_B @ lora_A).transpose(0,1) / alpha
    return beta_new
```

---

form an arithmetic set with the common difference being $\alpha_j$[1]. This is intractable in continuous and gradient-based optimization unless we ask that $c_{i,j}$ is a constant, *i.e.*, $c_{1,j} = \ldots = c_{i,j} = \ldots, c_{D_{\text{in}},j}$ for any $j$. This is equivalent to set all row vectors of $\mathbf{A}$ to be same, *i.e.*, $\mathbf{a}_1 \equiv \ldots \equiv \mathbf{a}_i \equiv \ldots \equiv \mathbf{a}_{D_{\text{in}}}$, where $\equiv$ denotes element-wise equivalence between two vectors.

The above strategy, while tractable, leads to a significant accuracy drop in practice. In particular, with all rows of $\mathbf{A}$ being the same vector, we have $\text{rank}(\mathbf{A}) = 1$ and thus $\text{rank}(\mathbf{AB}) = 1$, whereas the rank of $\mathbf{AB}$ is correlated to the ability of fine-tuning $\tilde{\mathbf{W}}$ in new data (Hu et al., 2021; Valipour et al., 2022; Dettmers et al., 2023a). To address this issue, a straightforward idea is to relax the constraints for both quantization and adaptation.

We partition each column of $\mathbf{W}$ into $L$ groups where, for ease of implementation, we set $L$ to be a divisor of $D_{\text{in}}$. Instead of quantizing each column of $\mathbf{W}$ entirely, we use an individual pair of scaling and zero factors for quantization, *i.e.*, the $l$-th group of factors, $\alpha_{l,j}$ and $\beta_{l,j}$, are computed for $D_{\text{in}}/L$ elements in the $j$-th column. Correspondingly, we only require the row vectors of $\mathbf{A}$ within the same group to have the same value. In our implementation, this is achieved by doing summation within each group of the input vector, $\mathbf{x}$. This parameter-free operation reduces the dimension of $\mathbf{x}$ from $D_{\text{in}}$ to $L$, hence we can set $\mathbf{A}$ to be a $L \times D_{\text{int}}$ matrix without further constraints.

The proposed approach is named quantization-aware low-rank adaptation (**QA-LoRA**). It is implemented by inserting/modifying a few lines of code beyond QLoRA, as shown in Algorithm 1[2]. Compared to LoRA, QA-LoRA enjoys advantages in time and memory consumption. Compared to QLoRA, QA-LoRA requires extra storage for $L \times D_{\text{out}}$ pairs of scaling and zero factors but reduces the number of parameters of $\mathbf{A}$ from $D_{\text{in}} \times D_{\text{int}}$ to $L \times D_{\text{int}}$ – since we often set $L \ll D_{\text{in}}$, the above change is negligible. Additionally, QA-LoRA is much faster in inference because it merges $s \cdot \mathbf{AB}$ into $\tilde{\mathbf{W}}$ while keeping the merged matrix $\mathbf{W}'$ quantized in low bits.

**The insight of QA-LoRA: balance.** QA-LoRA is very similar to a variant of QLoRA in which NF4 quantization is replaced by INT4[3]). In this version, the number of parameters for quantization ($D_{\text{out}}$ pairs of scaling and zero factors) is much smaller than that for adaptation ($D_{\text{in}} \times D_{\text{int}} + D_{\text{int}} \times D_{\text{out}}$ parameters), *i.e.*, there is a significant imbalance here. We introduce group-wise operations, increasing the number of parameters for quantization from $D_{\text{out}}$ to $L \times D_{\text{out}}$, meanwhile decreasing that for adaptation from $D_{\text{in}} \times D_{\text{int}} + D_{\text{int}} \times D_{\text{out}}$ to $L \times D_{\text{int}} + D_{\text{int}} \times D_{\text{out}}$. As we shall see in experiments, a moderate $L$ can achieve satisfying accuracy of language understanding meanwhile preserving computational efficiency.

---

[1]The exact conditions are two-fold. For any $j$, there exists a new zero factor $\beta'_j$ and a set of integers $c_{i,j}$ so that $c_{i,j} = \alpha_j \times \hat{c}_{i,j} + \beta'_j$. Additionally, the difference between the minimum and maximum of $\hat{w}_{i,j} + \hat{c}_{i,j}$ is not greater than $2^B - 1$ so that the summed weights can still be quantized into $B$-bit integers.

[2]The `merge_with_quantization` function is called after the training procedure for merging weights.

[3]We implemented this version of QLoRA, and it reports very similar ($\pm 0.5\%$) accuracy compared to the original QLoRA in the few-shot experiments for MMLU.

Table 1: 0-shot and 5-shot accuracy (%) on the Massive Multitask Language Understanding (MMLU) dataset (Hendrycks et al., 2021). Each block is based on the same foundation model specified at the first row. We organize all results using the fine-tuning dataset (Alpaca or Flan-v2) and the bit width of quantization. The bit width of '4 + 16' refers to the original QLoRA where the final version for inference is in FP16. Some important numbers are plotted in Figures 4–7 (please see Appendix B).

| Method | Dataset | #Bits | MMLU (0-shot) | | | | | MMLU (5-shot) | | | | |
|---|---|---|---|---|---|---|---|---|---|---|---|---|
| | | | Hums. | STEM | Social | Other | Avg. | Hums. | STEM | Social | Other | Avg. |
| LLaMA-7B | – | 16 | 32.4 | 26.6 | 31.4 | 37.2 | 32.1 | 33.3 | 29.8 | 37.8 | 38.0 | 34.6 |
| *QLoRA* | *Alpaca* | *4+16* | *38.1* | *31.1* | *41.6* | *46.9* | *39.4* | *36.1* | *31.9* | *42.0* | *44.5* | *38.4* |
| QLoRA w/ GPTQ | Alpaca | 4 | 35.7 | 30.9 | 38.0 | 44.0 | 37.1 | 33.8 | 31.3 | 37.4 | 42.2 | 36.0 |
| PEQA | Alpaca | 4 | – | – | – | – | – | 34.9 | 28.9 | 37.5 | 40.1 | 34.8 |
| **QA-LoRA** | Alpaca | 4 | 36.9 | 31.4 | 40.3 | 44.9 | **38.3** | 36.6 | 32.4 | 44.8 | 44.9 | **39.4** |
| QLoRA w/ GPTQ | Alpaca | 3 | 31.5 | 28.9 | 31.8 | 36.8 | 32.2 | 31.6 | 30.1 | 35.6 | 39.8 | 34.0 |
| **QA-LoRA** | Alpaca | 3 | 36.0 | 34.1 | 42.0 | 42.3 | **38.3** | 35.6 | 30.5 | 41.5 | 42.7 | **37.4** |
| QLoRA w/ GPTQ | Alpaca | 2 | 24.1 | 22.1 | 22.5 | 23.7 | 23.2 | 23.4 | 26.2 | 26.4 | 28.4 | 25.8 |
| **QA-LoRA** | Alpaca | 2 | 26.4 | 25.5 | 25.6 | 28.7 | **26.5** | 27.3 | 26.1 | 26.1 | 30.3 | **27.5** |
| *QLoRA* | *FLAN v2* | *4+16* | *40.9* | *32.5* | *47.8* | *49.5* | *42.6* | *41.4* | *35.0* | *49.8* | *52.0* | *44.3* |
| QLoRA w/ GPTQ | FLAN v2 | 4 | 39.7 | 32.5 | 46.4 | 48.1 | 41.6 | 36.5 | 33.7 | 46.9 | 50.3 | 41.4 |
| **QA-LoRA** | FLAN v2 | 4 | 44.0 | 35.3 | 52.3 | 52.6 | **45.9** | 43.9 | 38.0 | 54.3 | 53.0 | **47.0** |
| QLoRA w/ GPTQ | FLAN v2 | 3 | 36.7 | 30.2 | 38.4 | 40.1 | 36.5 | 32.2 | 31.7 | 42.7 | 42.8 | 36.9 |
| **QA-LoRA** | FLAN v2 | 3 | 41.4 | 35.1 | 52.0 | 50.2 | **44.4** | 41.3 | 36.0 | 52.8 | 50.2 | **44.7** |
| QLoRA w/ GPTQ | FLAN v2 | 2 | 24.1 | 22.5 | 22.3 | 23.8 | 23.3 | 23.9 | 25.3 | 26.2 | 25.3 | 25.0 |
| **QA-LoRA** | FLAN v2 | 2 | 34.1 | 30.0 | 37.2 | 39.8 | **35.2** | 31.8 | 38.1 | 34.5 | 38.5 | **33.2** |
| LLaMA-13B | – | 16 | 40.6 | 36.7 | 48.9 | 48.0 | 43.3 | 44.0 | 35.9 | 53.2 | 52.9 | 46.3 |
| *QLoRA* | *Alpaca* | *4+16* | *45.2* | *38.3* | *55.0* | *54.6* | *48.1* | *46.0* | *37.3* | *55.8* | *55.1* | *48.4* |
| QLoRA w/ GPTQ | Alpaca | 4 | 44.7 | 38.0 | 54.4 | 54.0 | 47.6 | 45.4 | 37.4 | 55.7 | 54.3 | 48.0 |
| PEQA | Alpaca | 4 | – | – | – | – | – | 43.0 | 37.7 | 53.6 | 49.0 | 45.0 |
| **QA-LoRA** | Alpaca | 4 | 44.3 | 38.0 | 55.1 | 55.5 | **47.9** | 48.4 | 38.3 | 54.9 | 55.2 | **49.2** |
| QLoRA w/ GPTQ | Alpaca | 3 | 43.5 | 36.2 | 52.3 | 52.6 | 45.9 | 43.6 | 36.1 | 53.0 | 52.7 | 46.1 |
| **QA-LoRA** | Alpaca | 3 | 43.9 | 37.3 | 53.1 | 54.3 | **46.9** | 44.3 | 38.8 | 53.4 | 53.8 | **47.3** |
| QLoRA w/ GPTQ | Alpaca | 2 | 27.7 | 27.6 | 31.8 | 29.7 | 29.0 | 29.0 | 27.1 | 33.4 | 34.8 | 30.9 |
| **QA-LoRA** | Alpaca | 2 | 35.7 | 33.3 | 40.9 | 42.0 | **37.8** | 35.6 | 30.6 | 39.9 | 41.7 | **36.9** |
| *QLoRA* | *FLAN v2* | *4+16* | *48.0* | *39.2* | *58.2* | *56.7* | *50.3* | *49.9* | *40.1* | *60.2* | *57.9* | *51.9* |
| QLoRA w/ GPTQ | FLAN v2 | 4 | 47.6 | 39.6 | 57.6 | 56.0 | 50.0 | 49.4 | 40.9 | 59.7 | 57.6 | 51.7 |
| **QA-LoRA** | FLAN v2 | 4 | 47.7 | 41.4 | 59.6 | 57.2 | **51.1** | 50.0 | 41.5 | 60.5 | 58.4 | **52.4** |
| QLoRA w/ GPTQ | FLAN v2 | 3 | 46.6 | 37.9 | 55.9 | 55.7 | 48.9 | 46.5 | 38.2 | 57.2 | 56.1 | 49.3 |
| **QA-LoRA** | FLAN v2 | 3 | 47.4 | 39.4 | 57.7 | 56.0 | **49.9** | 49.3 | 40.0 | 60.0 | 57.5 | **51.5** |
| QLoRA w/ GPTQ | FLAN v2 | 2 | 36.2 | 30.3 | 40.8 | 44.1 | 37.8 | 36.6 | 32.0 | 43.8 | 44.2 | 38.9 |
| **QA-LoRA** | FLAN v2 | 2 | 40.8 | 36.4 | 39.3 | 50.1 | **43.9** | 40.9 | 36.1 | 50.7 | 46.7 | **44.1** |
| LLaMA-33B | – | 16 | 51.0 | 42.7 | 63.3 | 60.4 | 54.1 | 56.2 | 45.9 | 67.1 | 63.9 | 58.2 |
| *QLoRA* | *Alpaca* | *4+16* | *52.2* | *44.9* | *64.3* | *61.8* | *55.5* | *55.4* | *46.0* | *66.4* | *63.6* | *57.7* |
| QLoRA w/ GPTQ | Alpaca | 4 | 51.7 | 44.7 | 63.4 | 61.0 | 54.9 | 53.9 | 46.6 | 66.3 | 62.9 | 57.1 |
| **QA-LoRA** | Alpaca | 4 | 51.6 | 44.9 | 65.0 | 61.8 | **55.4** | 55.8 | 46.4 | 67.0 | 64.0 | **58.1** |
| QLoRA w/ GPTQ | Alpaca | 3 | 49.5 | 43.3 | 63.1 | 61.0 | 53.8 | 53.3 | 45.0 | 64.1 | 61.4 | 55.8 |
| **QA-LoRA** | Alpaca | 3 | 50.6 | 44.6 | 64.0 | 61.2 | **54.7** | 54.3 | 45.8 | 65.2 | 62.6 | **56.8** |
| QLoRA w/ GPTQ | Alpaca | 2 | 32.0 | 31.6 | 35.8 | 32.8 | 32.9 | 37.5 | 34.9 | 45.3 | 44.9 | 40.4 |
| **QA-LoRA** | Alpaca | 2 | 38.4 | 38.2 | 50.7 | 49.7 | **43.6** | 44.2 | 38.8 | 53.9 | 52.3 | **47.0** |
| *QLoRA* | *FLAN v2* | *4+16* | *56.3* | *46.5* | *68.6* | *64.6* | *58.8* | *57.2* | *48.6* | *69.8* | *65.2* | *60.0* |
| QLoRA w/ GPTQ | FLAN v2 | 4 | 54.9 | 46.4 | 68.2 | 63.6 | 58.0 | 57.4 | 48.6 | 69.2 | 64.9 | 59.8 |
| **QA-LoRA** | FLAN v2 | 4 | 54.2 | 47.0 | 69.7 | 65.5 | **58.7** | 57.9 | 48.8 | 71.0 | 65.5 | **60.6** |
| QLoRA w/ GPTQ | FLAN v2 | 3 | 54.0 | 44.3 | 65.8 | 62.7 | 56.5 | 55.7 | 47.4 | 67.9 | 64.0 | 58.5 |
| **QA-LoRA** | FLAN v2 | 3 | 53.1 | 45.0 | 66.9 | 63.0 | **56.7** | 56.8 | 46.9 | 68.9 | 63.7 | **58.9** |
| QLoRA w/ GPTQ | FLAN v2 | 2 | 37.9 | 35.0 | 47.6 | 42.9 | 40.6 | 42.8 | 37.0 | 54.3 | 51.5 | 46.1 |
| **QA-LoRA** | FLAN v2 | 2 | 49.4 | 40.4 | 59.8 | 56.5 | **51.4** | 49.6 | 42.7 | 60.7 | 57.8 | **52.4** |
| LLaMA-65B | – | 16 | 56.4 | 45.2 | 68.0 | 64.1 | 58.3 | 61.4 | 51.9 | 73.6 | 67.6 | 63.4 |
| *QLoRA* | *Alpaca* | *4+16* | *55.5* | *49.3* | *70.4* | *66.9* | *60.1* | *60.3* | *52.7* | *72.9* | *67.4* | *63.1* |
| QLoRA w/ GPTQ | Alpaca | 4 | 54.8 | 48.9 | 69.8 | 66.1 | 59.4 | 60.4 | 52.5 | 73.0 | 67.2 | **63.0** |
| **QA-LoRA** | Alpaca | 4 | 57.1 | 48.2 | 70.7 | 64.9 | **60.0** | 60.8 | 50.5 | 72.5 | 66.7 | 62.5 |
| QLoRA w/ GPTQ | Alpaca | 3 | 57.4 | 47.9 | 67.2 | 65.1 | 59.3 | 59.6 | 50.0 | 70.6 | 66.1 | 61.4 |
| **QA-LoRA** | Alpaca | 3 | 57.6 | 48.4 | 69.3 | 65.4 | **60.0** | 59.3 | 49.6 | 71.9 | 66.0 | **61.5** |
| QLoRA w/ GPTQ | Alpaca | 2 | 43.9 | 38.0 | 42.6 | 51.1 | 46.2 | 47.3 | 40.8 | 58.9 | 57.0 | 50.7 |
| **QA-LoRA** | Alpaca | 2 | 48.6 | 42.5 | 60.7 | 58.6 | **52.2** | 51.3 | 43.4 | 63.4 | 60.7 | **54.4** |
| *QLoRA* | *FLAN v2* | *4+16* | *58.8* | *52.5* | *74.0* | *67.4* | *62.8* | *59.8* | *52.9* | *75.0* | *69.6* | *63.9* |
| QLoRA w/ GPTQ | FLAN v2 | 4 | 57.8 | 51.9 | 73.5 | 67.8 | 62.3 | 59.2 | 52.5 | 75.0 | 69.3 | **63.5** |
| **QA-LoRA** | FLAN v2 | 4 | 64.1 | 52.6 | 74.8 | 69.1 | **65.1** | 57.6 | 51.1 | 73.9 | 67.4 | 62.1 |
| QLoRA w/ GPTQ | FLAN v2 | 3 | 58.5 | 50.2 | 71.5 | 66.9 | **61.5** | 59.9 | 51.7 | 73.4 | 67.9 | 63.0 |
| **QA-LoRA** | FLAN v2 | 3 | 57.5 | 49.5 | 72.4 | 66.9 | 61.2 | 61.7 | 51.1 | 73.8 | 68.4 | **63.6** |
| QLoRA w/ GPTQ | FLAN v2 | 2 | 47.9 | 43.1 | 60.1 | 56.0 | 51.4 | 52.6 | 43.8 | 62.8 | 58.5 | 54.3 |
| **QA-LoRA** | FLAN v2 | 2 | 55.9 | 44.6 | 65.6 | 63.4 | **57.1** | 55.5 | 46.8 | 67.3 | 63.2 | **58.0** |

## 4 EXPERIMENTS

### 4.1 SETTINGS

**Foundation models.** We establish QA-LoRA upon the LLaMA (Touvron et al., 2023a) and LLaMa2 (Touvron et al., 2023b) families. In particular, we fine-tune the 7B, 13B, 33B, and 65B models of LLaMA and the 7B and 13B models of LLaMA2.

**Evaluation metrics.** Following QLoRA (Dettmers et al., 2023a), we evaluate both the zero-shot and few-shot performance of the LLMs on Massively Multitask Language Understanding (MMLU) benchmark (Hendrycks et al., 2021). It consists of 57 language tasks including humanities, STEM, social science, *etc.* We use the official MMLU evaluation script and prompts[4]. We further assess the zero-shot common sense reasoning ability on tasks covering HellaSwag (Zellers et al., 2019), PIQA (Bisk et al., 2020), WinoGrande (Sakaguchi et al., 2019), ARC (Clark et al., 2018), BoolQ (Clark et al., 2019), and OpenBookQA (Mihaylov et al., 2018). We adopt lm-eval-harness (Gao et al., 2021) to produce the Common Sense QA results.

**Quantization.** We adopt GPTQ (Frantar et al., 2023) in the quantization step, and our approach is open to other PTQ methods such as (Lin et al., 2023; Dettmers et al., 2023b). We use the same settings to quantize the QLoRA fine-tuned models and pre-trained LLaMA models. In the main experiments, we conduct a group-wise asymmetric quantization (with a group size of 32). We set the `act-order` variable to be `false` and the `true-sequential` variable to be `true`.

**Datasets and training details.** We choose Alpaca (Taori et al., 2023) and FLAN v2 (Longpre et al., 2023) as our fine-tuning datasets. Alpaca contains 52K instruction-following data generated from text-davinci-003 (GPT 3.5) (Wang et al., 2022). FLAN v2 is a collection of 1,836 tasks combining the mixture with CoT, Muffin, T0-SF, and NIV2. To save the tuning cost, we randomly sample a 320K subset from the FLAN v2 collection. Following QLoRA (Dettmers et al., 2023a), we use a paged AdamW optimizer, a maximum gradient norm of 0.3, and a batch size of 16 in the tuning period. We choose the constant learning rate schedule and set the learning rate to be $2 \times 10^{-5}$ for the 7B and 13B models and $1 \times 10^{-5}$ for the 33B and 65B models. The number of fine-tuning steps is 10K for Alpaca and 20K for FLAN v2. All experiments are conducted on Tesla V100 GPUs. We use one GPU for the 7B, 13B, and 33B models and two GPUs for the 65B models.

### 4.2 MAIN RESULTS AND EFFICIENCY

**Comparison against recent competitors on LLaMA for MMLU.** We first apply QA-LoRA to fine-tune the LLaMA models for MMLU. Table 1 summarizes the results with respect to different model sizes, fine-tuning datasets, and bit widths. Besides the base LLaMA models, we also compare QA-LoRA against QLoRA (Dettmers et al., 2023a), the most related work, and PEQA (Kim et al., 2023), a recent quantization method that does not use LoRA. We report both the original QLoRA (the inference stage involves FP16 computation) and the variant after GPTQ (for fair comparison). QA-LoRA consistently outperforms both competitors (QLoRA w/ GPTQ and PEQA) in either 0-shot and 5-shot accuracy. The advantage is more significant when the model size is small (*e.g.*, 7B and 13B) or the bit width is small (*e.g.*, INT3 or even INT2 is used), demonstrating that QA-LoRA is a strong solution in the scenarios that require computational efficiency. In some cases, the INT4 version of QA-LoRA performs even better than the original version of QLoRA meanwhile the inference speed is much faster (see the next paragraph). We further demonstrate some examples of QA-LoRA in Appendix A, where one can see the qualitative comparison and QA-LoRA beyond QLoRA w/ GPTQ. QA-LoRA mainly benefits from the quantization-aware adaptation; otherwise, the post-training quantization will not be compensated, resulting in unstable results.

**Application to large models.** As shown in Table 1, in large models (*e.g.*, 33B and 65B), QA-LoRA still achieves improvement over the baseline, QLoRA with GPTQ. Upon these results, we discuss the need for QA-LoRA in large models in the following aspects. (1) QA-LoRA reduces the computational costs of fine-tuning large models, *e.g.*, using QA-LoRA, only 1 and 2 V100 GPUs are needed for fine-tuning the 33B and 65B models. (2) When larger models are used, there can be increasing needs for low-bit (*e.g.*, INT3 and INT2) quantization, especially when the large models are to be deployed to edge devices. QA-LoRA shows significant advantages in such scenarios.

---

[4] https://github.com/hendrycks/test

Table 2: The numbers of learnable parameters and time costs of QLoRA and QA-LoRA during the fine-tuning stage. All results are reported on Alpaca with one Tesla-V100 GPU (the 65B model uses two chips). The number of fine-tuning steps is 10K.

| Method | LLaMA-7B | | LLaMA-13B | | LLaMA-33B | | LLaMA-65B | |
| | #Params | Time$_{(h)}$ | #Params | Time$_{(h)}$ | #Params | Time$_{(h)}$ | #Params | Time$_{(h)}$ |
|---|---|---|---|---|---|---|---|---|
| QLoRA | 160M | 40.0 | 250M | 73.1 | 488M | 148.6 | 800M | 284.5 |
| **QA-LoRA** | 89M | **21.5** | 140M | **29.5** | 272M | **51.2** | 447M | **100.5** |
| **QA-LoRA** | 178M | 21.8 | 280M | 30.1 | 544M | 52.2 | 894M | 103.5 |

Table 3: 0-shot commonsense QA accuracy (%) with respect to different quantization bit widths.

| Method | #Bits | HellaSwag | PIQA | WinoGrande | ARC-e | ARC-c | BoolQ | OBQA | Avg. |
|---|---|---|---|---|---|---|---|---|---|
| LLaMA-7B | 16 | 56.3 | 78.2 | 67.1 | 67.3 | 38.2 | 72.9 | 28.4 | 58.3 |
| *QLoRA* | *4+16* | *61.8* | *78.1* | *68.4* | *75.8* | *43.6* | *73.7* | *32.8* | *62.0* |
| LLaMA-7B + GPTQ | 4 | 54.5 | 76.5 | 66.9 | 66.1 | 36.9 | 70.9 | 27.4 | 57.0 |
| QLoRA w/ GPTQ | 4 | 57.4 | 77.6 | 66.2 | 70.9 | 41.8 | 73.5 | 31.2 | 59.8 |
| **QA-LoRA** | 4 | 58.6 | 78.0 | 66.9 | 71.2 | 43.9 | 79.9 | 34.0 | 61.8 |
| QLoRA w/ GPTQ | 3 | 52.2 | 75.2 | 64.1 | 65.8 | 37.2 | 70.4 | 27.2 | 56.0 |
| **QA-LoRA** | 3 | 57.6 | 76.2 | 66.5 | 70.2 | 43.1 | 76.3 | 30.6 | 60.1 |
| QLoRA w/ GPTQ | 2 | 31.9 | 58.2 | 52.4 | 32.3 | 20.7 | 60.6 | 14.6 | 38.7 |
| **QA-LoRA** | 2 | 49.8 | 70.2 | 58.5 | 55.4 | 33.9 | 73.7 | 32.8 | 53.7 |

Table 4: 0-shot and 5-shot MMLU accuracy (%) based on the LLaMA2 model family.

| Method | Data | #Bits | MMLU (0-shot) | | | | | MMLU (5-shot) | | | | |
| | | | Hums. | STEM | Social | Other | Avg. | Hums. | STEM | Social | Other | Avg. |
|---|---|---|---|---|---|---|---|---|---|---|---|---|
| LLaMA2-7B | – | 16 | 38.9 | 32.9 | 46.6 | 44.9 | 40.7 | 43.0 | 36.4 | 51.4 | 52.2 | 45.5 |
| **QA-LoRA** | Alpaca | 4 | 41.1 | 35.4 | 50.2 | 50.1 | 43.9 | 42.1 | 34.4 | 49.1 | 50.3 | 43.9 |
| **QA-LoRA** | FLAN v2 | 4 | 47.4 | 39.5 | 58.9 | 57.3 | **50.5** | 48.4 | 41.4 | 59.4 | 58.6 | **51.7** |
| LLaMA2-13B | – | 16 | 48.1 | 42.7 | 60.5 | 59.5 | 52.3 | 53.3 | 44.1 | 63.3 | 61.0 | 55.3 |
| **QA-LoRA** | Alpaca | 4 | 48.2 | 41.7 | 60.4 | 58.7 | 51.9 | 48.0 | 43.0 | 59.7 | 57.4 | 51.7 |
| **QA-LoRA** | FLAN v2 | 4 | 50.7 | 44.1 | 63.8 | 62.0 | **54.8** | 52.9 | 44.8 | 65.9 | 64.0 | **56.6** |

**The efficiency of QA-LoRA.** A clear advantage of QA-LoRA lies in its computational efficiency. Table 2 compares QA-LoRA to QLoRA in terms of the learnable parameters and training time during the fine-tuning stage.

The reason behind the fewer amounts of parameters, compared to QLoRA, lies in the reduction of the dimensionality of $\mathbf{A}$. Compared to LoRA and QLoRA where $\mathbf{A}$ has $D_{in} \times D_{int}$ parameters, QA-LoRA reduces the number to $L \times D_{int}$ where $L$ is the group size and $L \ll D_{in}$. This reduces the number of parameters in QA-LoRA by around $1/2$ (originally, $\mathbf{A}$ and $\mathbf{B}$ have similar numbers of parameters). QA-LoRA achieves higher fine-tuning accuracy with fewer parameters.

Regarding the time cost of fine-tuning, it is not largely impacted by the parameters because the amount of LoRA parameters is much smaller than that of the LLM itself (*e.g.*, 89M or 160M *vs.* 7B). To verify this point, we double $D_{int}$ which also doubles the number of parameters, surpassing that of QLoRA, but QA-LoRA is still much faster than QLoRA (see the table below). The significant advantage of QA-LoRA in training time mainly comes from the use of INT4 quantization. Compared to NF4 quantization used by QLoRA, INT4 operators have been optimized by CUDA and are much faster in execution. Additionally, during the inference stage, QA-LoRA is also more than $50\%$ faster than QLoRA because the fine-tuned model (after weight integration) is still in INT4, unlike QLoRA that converts it back to FP16.

**Commonsense QA results.** We also evaluate QA-LoRA for 0-shot commonsense QA based on LLaMA-7B. Results are summarized in Table 3. Similar to the MMLU results, the 4-bit QA-LoRA is comparable with the mixed-precision QLoRA and outperforms the post-quantized QLoRA by an average of $2.0\%$. The advantage becomes more significant in low-bit scenarios, *e.g.*, the 2-bit QA-LoRA reports a remarkable accuracy gain of $15.0\%$ over the 2-bit post-quantized QLoRA.

**On LLaMA2 models.** We further validate the effectiveness of our method on LLaMA2 (Touvron et al., 2023b). As shown in Table 4, we fine-tune the 7B and 13B models of LLaMA2 and test them on MMLU. Compared to the original FP16 models, the INT4 models fine-tuned with FLAN v2 are consistently better, while those with Alpaca report slightly lower accuracy. These experiments validate that QA-LoRA is generalized to other pre-trained model families.

Table 5: 0-shot and 5-shot MMLU accuracy (%) on different fine-tuning datasets.

| Base Model | Method | #Bits | Self-instruct | | Longform | | Chip2 | | Alpaca | | Flan v2 | |
|---|---|---|---|---|---|---|---|---|---|---|---|---|
| | | | 0-shot | 5-shot | 0-shot | 5-shot | 0-shot | 5-shot | 0-shot | 5-shot | 0-shot | 5-shot |
| | *QLoRA* | *4+16* | *–* | *36.4* | *–* | *32.1* | *–* | *34.5* | *–* | *38.8* | *–* | *44.5* |
| LLaMA-7B | QLoRA w/ GPTQ | 4 | – | 35.4 | – | 29.3 | – | 33.6 | – | 36.0 | – | 41.4 |
| | **QA-LoRA** | 4 | 32.5 | 34.4 | 29.3 | 33.6 | 30.4 | 32.2 | 38.3 | 39.4 | 45.9 | 47.0 |
| | *QLoRA* | *4+16* | *–* | *39.0* | *–* | *43.2* | *–* | *41.6* | *–* | *48.4* | *–* | *51.9* |
| LLaMA-13B | QLoRA w/ GPTQ | 4 | – | 38.4 | – | 42.8 | – | 41.3 | – | 48.0 | – | 51.7 |
| | **QA-LoRA** | 4 | 44.4 | 46.1 | 39.9 | 43.3 | 42.4 | 45.8 | 47.9 | 49.2 | 51.1 | 52.4 |

## 4.3 ABLATIVE STUDIES

**Impact of the quantization group size.** We investigate different settings of $L$, the hyper-parameter that controls the numbers of parameters for both quantization and low-rank adaptation. Results are reported in Table 6 (see Appendix B), where group size (*i.e.*, $D_{in}/L$ is displayed instead of $L$). Recall that a larger $L$ (corresponding to a smaller group size) implies a larger number of parameters, *i.e.*, a smaller quantization loss, and a larger number of adaptation parameters. Meanwhile, it also requires a larger number of storage and computation, though negligible as long as $L \gg 1$. One can observe that a larger $L$ (*e.g.*, group size is 32) often leads to higher accuracy, and the advantage becomes more significant when the quantization bit width is small, implying that a larger quantization loss needs to be compensated by a larger number of parameters.

**Impact of $D_{int}$.** We diagnose the performance with respect to $D_{int}$ in Table 7 (see Appendix B). We find that the MMLU accuracy is not largely impacted by the value of $D_{int}$ unless it is too small.

**Impact of fine-tuning datasets.** We also evaluate QA-LoRA on more datasets such as Self-instruct (Wang et al., 2022), Longform (Köksal et al., 2023), and Chip2 (LAION, 2023). Results are summarized in Table 5. Compared to Alpaca and FLAN v2, these datasets are relatively small, and thus the fine-tuned models report a bit weaker accuracy on MMLU. Note that, with LLaMA-13B as the foundation model, QA-LoRA consistently outperforms QLoRA with mixed precision, meanwhile being much faster in the inference stage.

**Impact of the size of fine-tuning datasets.** Lastly, we evaluate QA-LoRA on different subsets of FLAN v2. The dataset size varies from 160K, 240K, 320K, 400K, and 480K. LLaMA-7B is used as the foundation model. As shown in Figure 3, low-bit quantization asks for more data, yet 320K is sufficient for both the INT2 and INT4 variants of QA-LoRA.

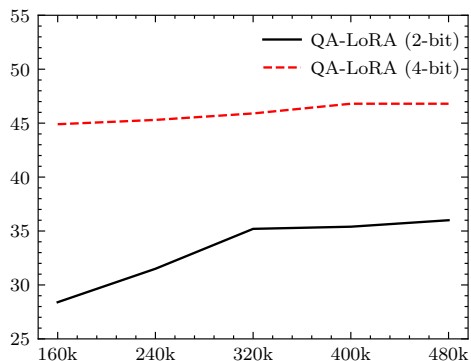

Figure 3: 5-shot MMLU accuracy (%) of QA-LoRA when the LLaMA-7B model is fine-tuned on subsets of FLAN v2 with different sizes.

## 5 CONCLUSION

In this paper, we propose **QA-LoRA** as an efficient method that introduces quantization awareness into the low-rank adaptation of LLMs. At the core of QA-LoRA lies the group-wise operations for both quantization and low-rank adaptation, and the key insight comes from balancing the numbers of parameters of both sides. QA-LoRA is easily implemented, generalized across various foundation models and language understanding tasks, and computationally efficient in both fine-tuning and inference stages. Extensive experiments on LLaMA validate the effectiveness of QA-LoRA.

## ETHICS STATEMENT

This paper is built upon pre-trained large language models (*e.g.*, LLaMA and LLaMA2) and existing datasets for instruct fine-tuning (*e.g.*, Alpaca and FLAN v2). We do not introduce any new data and thus do not involve human annotation. This paper has no additional ethical concerns beyond a large corpus of research in LLMs.

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

## A    QUALITATIVE STUDIES

In this section, we show a few examples of dialog. We compare our method, QA-LoRA, to the direct competitor, QLoRA, under different quantization bit widths. All QLoRA models are post-processed with GPTQ as described in the main text. We highlight inaccurate answers in blue and totally unacceptable answers in red.

Overall, QLoRA with GPTQ shows unstable behaviors across different cases. For example, in the second case, it crashes in 4-bit and 2-bit quantization but works well in 3-bit quantization. This mainly owes to the uncontrollable quantization loss in post-processing and such loss cannot be amended by any of the subsequent stages.

---

**Instruction**: Suggest five ways to budget your expenses.
**QLoRA** (4-**bit**): *1. Write down all of your current expenses. 2. Estimate your future expenses. 3. Cut down on non-essential spending. 4. Rate your expenses according to their importance. 5. Categorize your expenses to make it easier to.*
**Ours** (4-**bit**): *1. Plan a monthly budget and stick to it. 2. Track your spending. 3. Narrow down your wants and priorities. 4. Cut expenses by shopping around. 5. Eat at home as much as possible.*

———

**QLoRA** (3-**bit**): *1. Gather and document your financial information (income, expenses, and assets) 2. Estimate your cash flow (income, expenses, and assets) 3. Allocate your cash flow to the needed/wanted category. 4. Monitor your spending*
**Ours** (3-**bit**): *To budget expenses, one should: (1) keep track of spending; (2) have a spending plan; (3) set weekly goals; (4) only use cash; and (5) try to overspend one area to cover another.*

———

**QLoRA** (2-**bit**): *the maximum number of times 2026 #03289-012-052112.    03289-0222500032815-0451 2291-253222.*
**Ours** (2-**bit**): *1. Plan a monthly budget and stick to it. 2. Track your spending. 3. Narrow down your wants and priorities. 4. Cut expenses by shopping around. 5. Eat at home as much as possible.*

---

**Instruction**: How do you respond to a negative comment on social media?
**QLoRA** (4-**bit**): *I'm sorry you feel that way.*
**Ours** (4-**bit**): *Negative comments on social media can be hard to respond to, but I would first try to understand where the person is coming from and why they are saying what they are saying. I would then strive to reply in a way that does not further escalate the discussion, but instead responds with a mature.*

———

**QLoRA** (3-**bit**): *I would first thank the person for taking the time to share their thoughts. I would then say that I value their opinion and that I am here to listen and learn. I would then go on to explain why I feel the way I do and invite them to an open discussion.*
**Ours** (3-**bit**): *I response to a negative comment on social media by thanking the person for their feedback, clarifying if they are basing their comment off of a bad experience, and inviting them to contact me directly so we can work through any issues or concerns they may have.*

———

**QLoRA** (2-**bit**): *Demostimeta for the Parts of the Their work of for the parts of Their work of For the parts of3.4.2004 G Mach Rack 2002-2004 G 2.*
**Ours** (2-**bit**): *Negative comments on social media can be hard to respond to, but I would first try to understand where the person is coming from and why they are saying what they are saying. I would then strive to reply in a way that does not further escalate the discussion, but instead responds with a mature.*

---

**Instruction**: Categorize the following fruits according to color. Input: Apple, Banana, Avocado, Peach.
**QLoRA** (4-**bit**): *of the same color: Red (Apple, Banana), Green (Avocado, Peach) Other: Yellow (Peach).*

**Ours** (4-**bit**): *Apple - Red. Banana - Yellow. Avocado - Green. Peach - Pink.*
———————

**QLoRA** (3-**bit**): *is yellow is green is black is white.*
**Ours** (3-**bit**): *- Apple (Red)- Banana (Yellow)- Avocado (Green)- Peach (Pink).*
———————

**QLoRA** (2-**bit**): *Apple, mash up, and avocado.*
**Ours** (2-**bit**): *Apple - Red. Banana - Yellow. Avocado - Green. Peach - Pink.*

---

**Instruction**: Generate an HTML page that contains the text \"Welcome to my page\".
**QLoRA** (4-**bit**): *\n\n<html>\n <head>\n <title>Welcome to my page</title>\n </head>\n <body>\n <h1>Welcome to my page</h1>\n </body>\n</html>\n.*
**Ours** (4-**bit**): *Welcome to my page.\n<html>\n<body>\n<p>Welcome to my page.</p>\n</body>\n</html>\n.*
———————

**QLoRA** (3-**bit**): *<html>\n <head>\n <title>My Page</title>\n </head>\n <body>\n <p>Welcome to my page!</p>\n </body>\n</html> \n package org.michael.*
**Ours** (3-**bit**): *<HTML>\n <P>Welcome to my page</P>\n <P>Lorem Ipsum is simply placeholder text.</P>\n</HTML>\n\n Note that this response may also contain other elements such as stylesheets and script, if they were defined in the task.*
———————

**QLoRA** (2-**bit**): *The page has the text \"Welcome to my page\" as it.\n\n28. A reint: a 28!313473-020-520. The\n313473-020/2*
**Ours** (2-**bit**): *Welcome to my page.\n<html>\n<body>\n<p>Welcome to my page.</p>\n</body>\n</html>\n.*

---

**Instruction**: Create a dialogue between two people having a conversation about going to the beach.
**QLoRA** (4-**bit**): *of course, I'm in! what time were you thinking of meeting up? I was thinking of going around noon. sounds good, I'll see you then!*
**Ours** (4-**bit**): *.... (beach conversations A and B)\n.... B: Going to the beach sounds like a blast! A: It does, isn't it? B: Yep! A: so what are we doing? B: Planning a full itinerary.*
———————

**QLoRA** (3-**bit**): *"Person 1: "I'm so excited to go to the beach! The sun, sand, and water are all so soothing. I can't wait to get there and dive in!" Person 2: "Me too! I can't wait to experience that feeling of being".*
**Ours** (3-**bit**): *"So which beach should we go to, Fiji or Bora Bora?"\n, "Fiji has more coral reefs, Bora Bora has more beaches, right?"\n, "Yeah, Bora Bora also has the world's most"*
———————

**QLoRA** (2-**bit**): *Such as a user 31 is 4 0-5 Gigs Sector 4 (22.12.2018) - [P] / f / Beyond there is a lot of and.*
**Ours** (2-**bit**): *.... (beach conversations A and B)\n.... B: Going to the beach sounds like a blast! A: It does, isn't it? B: Yep! A: so what are we doing? B: Planning a full itinerary.*

## B  ADDITIONAL EXPERIMENTAL RESULTS

We plot some important results in Table 1 in Figures 4–7 to ease the readers to compare the our model with the baseline, QLoRA. Note that Figure 5 is identical to Figure 1.

We also provide the diagnosis on the $L$ (group size) and $D_{\text{int}}$ (the hidden dimensionality of adaptation) hyper-parameters in Tables 6 and 7.

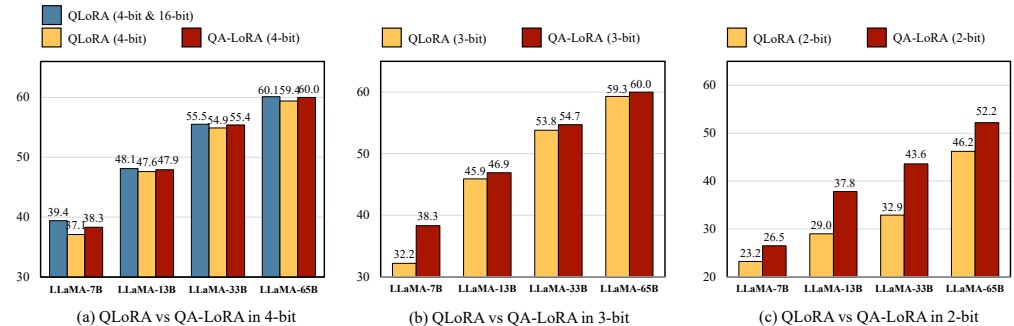

Figure 4: The comparison of 0-shot MMLU accuracy (%) with different quantization bit widths based on the LLaMA model family on the Alpaca dataset. Full results are provided in Table 1.

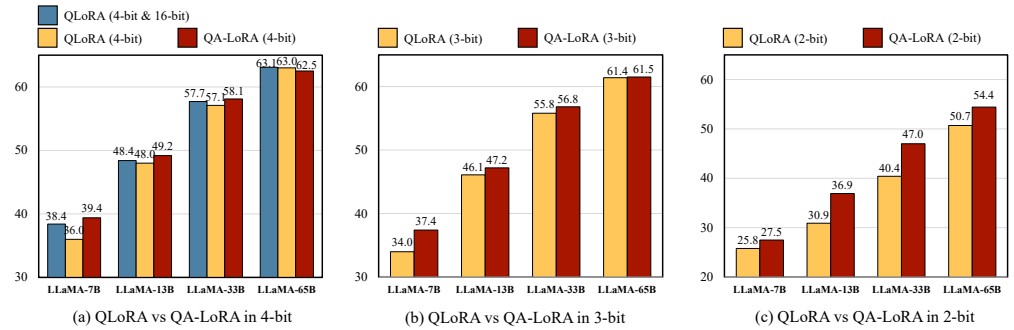

Figure 5: The comparison of 5-shot MMLU accuracy (%) with different quantization bit widths based on the LLaMA model family on the Alpaca dataset. Full results are provided in Table 1.

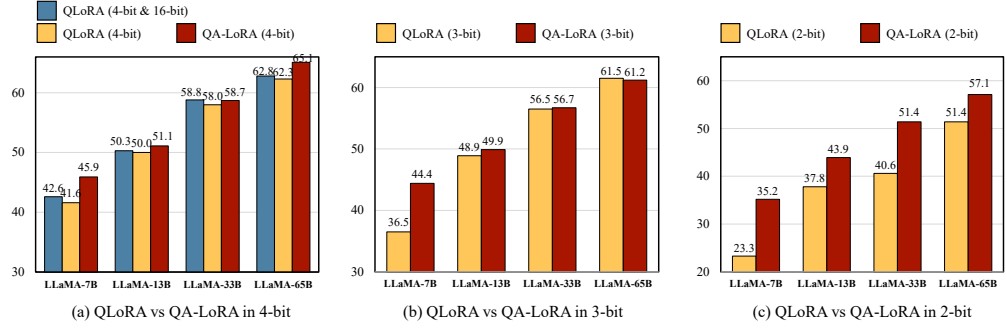

Figure 6: The comparison of 0-shot MMLU accuracy (%) with different quantization bit widths based on the LLaMA model family on the FLAN v2 dataset. Full results are provided in Table 1.

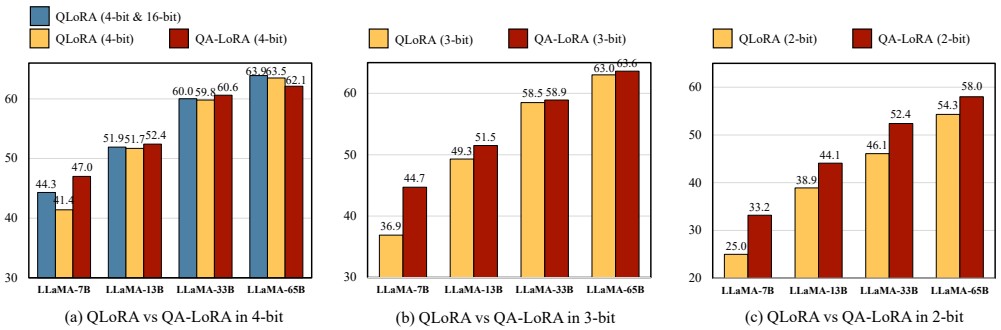

Figure 7: The comparison of 5-shot MMLU accuracy (%) with different quantization bit widths based on the LLaMA model family on the FLAN v2 dataset. Full results are provided in Table 1.

Table 6: 0-shot and 5-shot MMLU accuracy (%) on with respect to different group settings.

| Base Model | Group Size | #Bits | MMLU (0-shot) | | | | | MMLU (5-shot) | | | | |
|---|---|---|---|---|---|---|---|---|---|---|---|---|
| | | | Hums. (↑) | STEM (↑) | Social (↑) | Other (↑) | Avg. (↑) | Hums. (↑) | STEM (↑) | Social (↑) | Other (↑) | Avg. (↑) |
| LLaMA-7B | 128 | 4 | 37.3 | 31.8 | 39.3 | 43.7 | 38.0 | 36.5 | 32.1 | 41.7 | 44.0 | 38.4 |
| | 64 | 4 | 37.5 | 30.6 | 41.3 | 45.4 | 38.6 | 36.5 | 32.6 | 43.4 | 45.0 | **39.1** |
| | 32 | 4 | 38.1 | 31.1 | 41.6 | 46.9 | **39.4** | 36.1 | 31.9 | 42.0 | 44.5 | 38.4 |
| | 128 | 2 | 24.0 | 26.7 | 24.8 | 25.2 | 25.0 | 25.0 | 29.0 | 27.9 | 26.1 | 26.7 |
| | 64 | 2 | 25.1 | 26.9 | 24.7 | 27.0 | 25.8 | 25.0 | 27.2 | 25.2 | 27.3 | 26.0 |
| | 32 | 2 | 26.4 | 25.5 | 25.6 | 28.7 | **26.5** | 27.3 | 26.1 | 26.1 | 30.3 | **27.5** |
| LLaMA-13B | 128 | 4 | 43.4 | 39.6 | 55.5 | 53.9 | 47.6 | 46.5 | 38.0 | 55.8 | 54.5 | 48.6 |
| | 64 | 4 | 43.4 | 39.3 | 55.8 | 53.6 | 47.6 | 47.8 | 39.3 | 55.7 | 54.8 | **49.3** |
| | 32 | 4 | 44.3 | 38.0 | 55.1 | 55.5 | **47.9** | 48.4 | 38.3 | 54.9 | 55.2 | 49.2 |
| | 128 | 2 | 28.5 | 28.4 | 30.6 | 29.8 | 29.2 | 29.2 | 30.6 | 32.8 | 32.4 | 31.0 |
| | 64 | 2 | 30.7 | 31.5 | 38.1 | 36.0 | 33.7 | 32.3 | 30.3 | 37.0 | 38.3 | 34.3 |
| | 32 | 2 | 35.7 | 33.3 | 40.9 | 42.0 | **37.8** | 35.6 | 30.6 | 39.9 | 41.7 | **36.9** |

Table 7: 0-shot and 5-shot MMLU accuracy (%) by fine-tuning LLaMA-7B on the Alpaca dataset, with respect to different $D_{\text{int}}$ settings.

| Base Model | $D_{\text{int}}$ | #Bits | MMLU (0-shot) | | | | | MMLU (5-shot) | | | | |
|---|---|---|---|---|---|---|---|---|---|---|---|---|
| | | | Hums. (↑) | STEM (↑) | Social (↑) | Other (↑) | Avg. (↑) | Hums. (↑) | STEM (↑) | Social (↑) | Other (↑) | Avg. (↑) |
| LLaMA-7B | 1 | 4 | 35.6 | 31.5 | 39.0 | 43.6 | 37.3 | 35.1 | 31.4 | 43.1 | 44.0 | 38.1 |
| | 2 | 4 | 36.5 | 31.7 | 41.4 | 45.8 | 38.7 | 35.1 | 30.7 | 43.4 | 44.8 | 38.2 |
| | 4 | 4 | 37.3 | 32.5 | 41.0 | 45.7 | 39.0 | 36.6 | 32.3 | 44.8 | 44.3 | 39.3 |
| | 8 | 4 | 36.1 | 31.8 | 44.6 | 44.5 | 39.0 | 38.2 | 32.0 | 42.1 | 46.5 | 39.6 |
| | 16 | 4 | 37.2 | 32.3 | 41.3 | 45.5 | 39.0 | 36.8 | 31.8 | 45.2 | 45.4 | 39.5 |
| | 32 | 4 | 37.6 | 32.3 | 41.6 | 45.9 | 39.3 | 36.0 | 32.1 | 45.0 | 45.2 | 39.3 |
| | 64 | 4 | 37.7 | 31.9 | 41.7 | 45.1 | 39.0 | 36.4 | 31.8 | 45.0 | 44.8 | 39.2 |
| | 128 | 4 | 37.2 | 32.4 | 41.9 | 45.8 | 39.2 | 36.3 | 32.1 | 44.9 | 44.8 | 39.3 |
| | 1 | 2 | 27.4 | 24.4 | 25.4 | 27.4 | 26.3 | 27.1 | 25.4 | 26.0 | 28.2 | 26.7 |
| | 2 | 2 | 25.9 | 25.7 | 24.4 | 28.0 | 26.0 | 26.4 | 24.7 | 26.6 | 28.8 | 26.6 |
| | 4 | 2 | 26.7 | 25.7 | 25.3 | 28.6 | 26.6 | 26.2 | 25.0 | 26.6 | 29.3 | 26.7 |
| | 8 | 2 | 25.4 | 25.0 | 25.4 | 27.3 | 25.8 | 26.0 | 24.9 | 27.0 | 29.3 | 26.8 |
| | 16 | 2 | 25.5 | 24.7 | 24.5 | 28.7 | 25.9 | 26.5 | 24.7 | 26.0 | 29.5 | 26.7 |
| | 32 | 2 | 26.2 | 24.6 | 25.4 | 28.0 | 25.9 | 25.7 | 25.0 | 26.3 | 29.3 | 26.5 |
| | 64 | 2 | 26.4 | 23.7 | 24.0 | 26.4 | 25.3 | 27.5 | 26.3 | 26.4 | 28.5 | 27.2 |
| | 128 | 4 | 26.1 | 24.2 | 25.3 | 27.5 | 25.8 | 27.2 | 25.3 | 26.0 | 28.6 | 26.8 |

## C  MODEL SIZE

We report the sizes of the final models of QLoRA and QA-LoRA in Table 8. Please note that there are two ways for post-processing in QLoRA, *i.e.*, the unmerged ($\tilde{\mathbf{W}}$ and $s \cdot \mathbf{AB}$ are stored individually, which saves memory but the inference is slow) and merged ($s \cdot \mathbf{AB}$ is added to $\tilde{\mathbf{W}}$, which is faster in inference but requires large memory because the matrix must be stored in FP16). QA-LoRA enjoys both low memory usage and a fast inference speed. A side note: In 33B and 65B models, setting $L = 32$ in QA-LoRA results in slightly larger model sizes compared to QLoRA, but one can set $L = 128$ which causes a negligible accuracy drop.

Note that the final model size of QA-LoRA is exactly the size of $\mathbf{W}'$ (or equivalently, $\tilde{\mathbf{W}}$) because $s \cdot \mathbf{AB}$ is merged into $\tilde{\mathbf{W}}$ after adaptation. Take the 7B model with $L = 32$ as an example. The baseline, the unmerged version of QLoRA, is sized 4.6G, in which $\tilde{\mathbf{W}}$ is sized 4.0G and $\mathbf{A}$ and $\mathbf{B}$ combined is sized 0.6G. QA-LoRA increases the first amount to 4.3G and eliminates the second amount.

Table 8: The sizes (in GB) of the final models of QLoRA and QA-LoRA.

| Model | QLoRA (unmerged) | QLoRA (merged) | QA-LoRA ($B = 4$, $L = 32$) | QA-LoRA ($B = 4$, $L = 128$) |
|---|---|---|---|---|
| LLaMA-7B | 4.6 | 13.5 | 4.3 | 3.7 |
| LLaMA-13B | 8.1 | 24.4 | 8.1 | 6.9 |
| LLaMA-33B | 18.9 | 55.5 | 20.0 | 17.5 |
| LLaMA-65B | 36.1 | 122.3 | 39.0 | 34.7 |

