# OpenReview forum: "QA-LoRA: Quantization-Aware Low-Rank Adaptation of Large Language Models"
_ICLR.cc/2024/Conference — ICLR 2024 poster_

### Official Review · Reviewer_2aMS · 2023-10-17

**Soundness:** 4 excellent
**Presentation:** 3 good
**Contribution:** 3 good
**Rating:** 6
**Confidence:** 4

**Summary:**

This manuscript proposes QA-LoRA, a LoRA based parameter efficient LLM finetuning scheme with quantization. QA-LoRA extends QLoRA to be able to add low-rank matrices with pre-trained weights in low-bit tensors directly, without the need to PTQ on low-rank matrices. To guarantee that the summation of low-rank matrices and pre-trained weights are still within the same quantization range, the authors relax the requirement of each row being the same into groups, through group-wise quantization. This improves the accuracy and efficiency during inference. During evaluation, the authors experimented with a series of LLaMA and LLaMA2 models with different sizes. Results showed that the proposed models can achieve superior performance than LoRA and QLoRA.

**Strengths:**

* The paper organization, presentation, and references are good.
* The proposed method has enough novelty.

**Weaknesses:**

* Parameter offset in experiments: The proposed method incorporates group-wise/sub-channel qunatization, which includes an additional number of parameters for scales. Also, the proposed QA-LoRA reduces the size of low-rank matrices. However, these parameter offsets are not reflected in the results, which could be misleading to the audiences. It would be more informative to add the actual model size (or estimated) in MB/GB for each of the models.
* In the ablation study, only group size is examined. It would be worthwhile to experiment with the D_int as well, as it is also part of the tradeoff between model size and accuracy. It would be interesting to see what is the lowest D_int in this setup, compared to vanilla LoRA.

**Questions:**

See those in weaknesses.

---

> ### Author Response · Authors · 2023-11-22
> **Response to Reviewer 2aMS**
>
> We thank the reviewer for the comments and questions.
>
> >**Q1.** Parameter offset in experiments: The proposed method incorporates group-wise/sub-channel quantization, which includes an additional number of parameters for scales. Also, the proposed QA-LoRA reduces the size of low-rank matrices. However, these parameter offsets are not reflected in the results, which could be misleading to the audiences. It would be more informative to add the actual model size (or estimated) in MB/GB for each of the models.
>
> >**A1.** Thanks for the suggestion. First, we report the sizes of the final models of QLoRA and QA-LoRA in the following table. Please note that there are two ways for post-processing in QLoRA, *i.e.*, the unmerged ($\tilde{\mathbf{W}}$ and $s\cdot\mathbf{A}\mathbf{B}$ are stored individually, which saves memory but the inference is slow) and merged ($s\cdot\mathbf{A}\mathbf{B}$ is added to $\tilde{\mathbf{W}}$, which is faster in inference but requires large memory because the matrix must be stored in FP16). QA-LoRA enjoys both low memory usage and a fast inference speed. A side note: In 33B and 65B models, setting $L=32$ in QA-LoRA results in slightly larger model sizes compared to QLoRA, but one can set $L=128$ which causes a negligible accuracy drop.
> >Note that the final model size of QA-LoRA is exactly the size of $\mathbf{W}'$ (or equivalently, $\tilde{\mathbf{W}}$) because $s\cdot\mathbf{A}\mathbf{B}$ is merged into $\tilde{\mathbf{W}}$ after adaptation. Take the 7B model with $L=32$ as an example. The baseline, the unmerged version of QLoRA, is sized 4.6G, in which $\tilde{\mathbf{W}}$ is sized 4.0G and $\mathbf{A}$ and $\mathbf{B}$ combined is sized 0.6G. QA-LoRA increases the first amount to 4.3G and eliminates the second amount.
> >**In the revised paper, we have added the following tables to Appendix C.**
>
> |Models|QLoRA (unmerged)|QLoRA (merged)|QA-LoRA ($B=4$, $L=32$)|QA-LoRA ($B=4$, $L=128$)|
> |:-|:-:|:-:|:-:|:-:|
> |**LLaMA-7B**|4.6|13.5|4.3|3.7|
> |**LLaMA-13B**|8.1|24.4|8.1|6.9|
> |**LLaMA-33B**|18.9|55.5|20.0|17.5|
> |**LLaMA-65B**|36.1|122.3|39.0|34.7|
>
> Table. The model size (GB) of QLoRA and QA-LoRA with respect to different options. The QLoRA models used NF4 and FP16 numerics.
>
> >**Q2.** In the ablation study, only group size is examined. It would be worthwhile to experiment with the D_int as well, as it is also part of the tradeoff between model size and accuracy. It would be interesting to see what is the lowest D_int in this setup, compared to vanilla LoRA.
>
> >**A2.** Good suggestion! We follow QLoRA to set $D_\mathrm{int}=64$. During the rebuttal, we diagnose the performance with respect to $D_\mathrm{int}$ in the following table. We find that the MMLU accuracy is not largely impacted by the value of $D_\mathrm{int}$ unless it is too small (the same conclusion as in the QLoRA paper). We agree that discovering the smallest $D_\mathrm{int}$ value is interesting, and we empirically find that the smallest $D_\mathrm{int}$ that maintains the stability of QA-LoRA is $4$. **In the revised paper, we have added the table to Appendix B and the analysis to the main article.**
>
> |#Bits|D_int|MMLU (0-shot)|MMLU (5-shot)|#Bits|D_int|MMLU (0-shot)|MMLU (5-shot)|
> |:-|:-|:-:|:-:|:-|:-|:-:|:-:|
> |4|1|37.3|38.1|2|1|26.3|26.7|
> |4|2|38.7|38.2|2|2|26.0|26.6|
> |4|4|39.0|39.3|2|4|26.6|26.7|
> |4|8|39.0|39.6|2|8|25.8|26.8|
> |4|16|39.0|39.5|2|16|25.9|26.7|
> |4|32|39.3|39.3|2|32|25.9|26.5|
> |4|64|39.0|39.2|2|64|25.3|27.2|
> |4|128|39.2|39.3|2|128|25.8|26.8|
>
> Table. The MMLU accuracy (%) of $D_\mathrm{int}$ measured on INT4 and INT2 quantization upon LLaMA-7B on the Alpaca dataset.

---

### Official Review · Reviewer_YexA · 2023-10-31

**Soundness:** 4 excellent
**Presentation:** 4 excellent
**Contribution:** 3 good
**Rating:** 8
**Confidence:** 4

**Summary:**

The paper proposes a Quantization-Aware Low-Rank Adaptation (QA-LoRA) for efficient fine-tuning of LLM. This work comes improves Q-LORA algorithm by introducing group-wise operators which lift the need for post-training quantization. QA-LoRA implementation is simple and generic. It benefits from a balance between the number of parameters required for adaption and quantization. The experiments show that fine-tuning and inference stages are computationally efficient thanks to the use of INT4. The memory footprint of QA-LoRA is lower than QLoRA. In terms task accuracy, QA-LoRA is better than Q-LoRA with post-training quantization (GPTQ).

**Strengths:**

- This work solves a limitation of previous parameter-efficient tuning of LLMs by eliminating the need for a separate post-training quantization which drops model accuracy
- QA-LoRA further enhances memory efficiency of SOTA while preserving accuracy
- The experiments are convincing as they cover a wide range of scenarios

**Weaknesses:**

QA-LoRA introduce a hyper-parameter (L: Group size). This requires additional optimization and It is unclear if it can be selected without tuning.

**Questions:**

- I wonder if larger model where the need of this technique is crucial, e.g., 30B-60B, could be discussed.
- Figure 3 legend should have QA-LoRA instead of A-LoRA

---

> ### Author Response · Authors · 2023-11-22
> **Response to Reviewer YexA**
>
> We thank the reviewer for the comments and questions.
>
> >**Q1.** QA-LoRA introduce a hyper-parameter ($L$: group size). This requires additional optimization and it is unclear if it can be selected without tuning.
>
> >**A1.** Thanks for the question. We first analyze the impact of $L$. A smaller $L$ results in (1) a higher degree of freedom (which often leads to higher fine-tuning accuracy), (2) a larger memory to store the quantized weight matrix, $\tilde{\mathbf{W}}$, and (3) an increasing risk of over-fitting if $L$ is too small. Therefore, a good strategy is to set $L$ to be a relatively small (but not too small) integer, *e.g.*, $L=32$ as a good practice as shown in Table 5. We cannot guarantee that no hyper-parameter tuning is needed, but there is a clear guideline to do this. For example, one can try $L=16$, if the GPU memory allows, to achieve higher accuracy.
>
> >**Q2.** I wonder if a larger model where the need for this technique is crucial, e.g., 30B-60B, could be discussed.
>
> >**A2.** Nice suggestion! In the original version, we provide experiments on the 33B and 65B models of LLaMA in Table 1, and QA-LoRA achieves improvement over the baseline, QLoRA with GPTQ. Upon these results, we discuss the need for QA-LoRA in large models in the following aspects. (1) QA-LoRA reduces the computational costs of fine-tuning large models, *e.g.*, using QA-LoRA, only 1 and 2 V100 GPUs are needed for fine-tuning the 33B and 65B models. (2) When larger models are used, there can be increasing needs for low-bit (*e.g.*, INT3 and INT2) quantization, especially when the large models are to be deployed to edge devices. QA-LoRA shows significant advantages in such scenarios. **In the revised paper, the above contents are added to Section 4.2 where the experimental results are discussed.**
>
> >**Q3.** Figure 3 legend should have QA-LoRA instead of A-LoRA.
>
> >**A3.** Sorry for our carelessness. **It has been fixed in the revised version.**

---

### Official Review · Reviewer_KGp3 · 2023-11-06

**Soundness:** 3 good
**Presentation:** 2 fair
**Contribution:** 2 fair
**Rating:** 5
**Confidence:** 4

**Summary:**

This paper introduces a modification to the QLoRA method, designed to facilitate the training of large language models with limited computational resources. QLoRA initially quantizes neural network weights to NF4 format and subsequently optimizes LoRA matrix weights in FP16. This process increases inference latency, as everything is converted to FP16 during inference. The proposed alternative method outlined in this paper ensures the appropriate quantization of LoRA weights without necessitating a reversion to FP16 during inference. This enhancement involves just a few lines of code, offering a more efficient solution. While the paper lacks specific results and comprehensive explanations, it presents a promising direction for optimizing large language model training within constrained computational budgets.

**Strengths:**

**Addresses a Significant Issue** - QLoRA's potential is realized through its ability to quantize LoRA weights, effectively resolving the disparities observed between fine-tuning and inference in QLoRA.

**Streamlined Implementation** - The authors highlight the method's simplicity, emphasizing that it necessitates a mere two lines of code modification to yield impressive enhancements.

**Thorough Assessment** - The evaluation is meticulous, with the authors examining a spectrum of competitive methods and diverse model architectures to demonstrate the method's advantages comprehensively.

**Weaknesses:**

Reasoning behind the method - Wy should all the c_ij as defined in the paper be equal is not clear — which is the main motivation for the group-wise quantisation. I would be willing to improve the scores with better explanation on the explanation of the method (See the questions)

**Questions:**

1. In Algorithm 1, the function `merge_with_quantization` is defined but never used.
2. What is the degree of freedom of quantisation and adaptation - These seem to be new terms that are added in this paper and not used in the literature. These have to be defined, before claiming that they are increased or managed by the proposed method
3. Page 6 is just results - Page 6 is just results without much to interpret. These are a bunch of numbers. Please consider presenting this table in a better manner. Can this be presented as a graph for readers? Just numbers are hard to read
4. In Table 2, you indicate that the number of parameters in the method are lesser than QLoRA — almost by 2x. Why is this the case? This seems like a unfair comparison to QLoRA. What is the time taken in hours for fine-tuning with similar number of parameters.
5. Section 3.3 explains the method and the reasoning behind on why the rank degenerates to 1. But the explanation is not comprehensive.

---

> ### Author Response · Authors · 2023-11-22
> **Response to Reviewer KGp3 (Part 1)**
>
> We thank the reviewer for the comments and questions.
>
> >**Q0.** Reasoning behind the method - Why should all the c_ij as defined in the paper be equal is not clear — which is the main motivation for the group-wise quantization.
>
> >**A0.** Please refer to our response to **Q5**.
>
> >**Q1.** In Algorithm 1, the function `merge_with_quantization` is defined but never used.
>
> >**A1.** Sorry for misleading. Due to the space limit, we did not add the `main` function in the paper, which calls `qalora_training` (which calls `qalora_forward`) and `merge_with_quantization`. The `merge_with_quantization` function is called *after* the entire QA-LoRA training procedure for merging the LoRA weights $s\cdot\mathbf{A}\mathbf{B}$ into the quantized weight matrix $\tilde{\mathbf{W}}$. It is done by updating the $\beta$ parameter into $\beta_\mathrm{new}$. **In the revised paper, we have added an explanation in the main article, after the reference to Algorithm 1.**
>
> >**Q2.** What is the degree of freedom of quantization and adaptation - These seem to be new terms that are added in this paper and not used in the literature. These have to be defined, before claiming that they are increased or managed by the proposed method.
>
> >**A2.** Thanks for the question. Literally, the degree of freedom (DoF) means how many parameters (or pairs of parameters) can be used for quantization or adaptation. We explain our insight as follows and, **to avoid confusion, we replaced all "the degree of freedom" with "the number of parameters" in the revised paper.**
> >In QLoRA, each column of $\mathbf{W}$ is quantized using the same pair of $\alpha$ and $\beta$ parameters, which we say the DoF (number of parameters) of the *quantization* part is $1$; meanwhile, the matrix $\mathbf{A}$ is sized $D_\mathrm{in}\times D_\mathrm{int}$, implying that each of the $D_\mathrm{in}$ entries in the column of $\tilde{\mathbf{W}}$ is multiplied by an individual parameter, which we say the DoF (number of parameters) of the *adaptation* part is $D_\mathrm{in}$. There is clearly an imbalance here, which causes two issues: (1) the over-high quantization error harms the fine-tuning accuracy, and (2) the adaptation weights cannot be merged to the quantized matrix, $\tilde{\mathbf{W}}$.
> >This motivates us to (1) increase the number of parameters of the *quantization* part from $1$ to $D_\mathrm{in}/L$ using group-wise quantization, where $L$ is the group size, and (2) decrease the number of parameters of the *adaptation* part by squeezing the input dimensionality of $\mathbf{A}$ from $D_\mathrm{in}$ to $L$. This is why QA-LoRA improves the accuracy and naturally allows the adaptation weights to be merged into the quantized matrix.
>
> >**Q3.** Page 6 is just results - Page 6 is just results without much to interpret. These are a bunch of numbers. Please consider presenting this table in a better manner. Can this be presented as a graph for readers? Just numbers are hard to read.
>
> >**A3.** Thanks for the suggestion. We followed QLoRA and other recent works to report these exact numbers to ease the comparison against existing papers. Actually, the main results of Table 6 (5-shot MMLU on the Alpaca dataset) have been summarized in Figure 1. **In the revised paper, we plot other results (0-shot MMLU and the FLAN v2 dataset) as similar figures and add them to Appendix B.**
>
> =====*to be continued in Part 2*=====

---

> ### Author Response · Authors · 2023-11-22
> **Response to Reviewer KGp3 (Part 2)**
>
> =====*continuing from Part 1*=====
>
>
> >**Q4.** In Table 2, you indicate that the number of parameters in the method is lesser than QLoRA — almost by 2x. Why is this the case? This seems like an unfair comparison to QLoRA. What is the time taken in hours for fine-tuning with a similar number of parameters?
>
> >**A4.** The reason behind the fewer amounts of parameters, compared to QLoRA, lies in the reduction of the dimensionality of $\mathbf{A}$. Compared to LoRA and QLoRA where $\mathbf{A}$ has $D_\mathrm{in}\times D_\mathrm{int}$ parameters, QA-LoRA reduces the number to $L\times D_\mathrm{int}$ where $L$ is the group size and $L\ll D_\mathrm{in}$. This reduces the number of parameters in QA-LoRA by around $1/2$ (originally, $\mathbf{A}$ and $\mathbf{B}$ have similar numbers of parameters). We would like to stress that this is a good feature of QA-LoRA which allows it to achieve higher fine-tuning accuracy with fewer parameters.
> >Regarding the fine-tuning time, it is *not* largely impacted by the parameters because the amount of LoRA parameters is much smaller than that of the LLM itself (*e.g.*, 89M or 160M *vs.* 7B). To verify this point, we double $D_\mathrm{int}$ which also doubles the number of parameters, surpassing that of QLoRA, but QA-LoRA is still much faster than QLoRA (see the table below). Essentially, QA-LoRA is faster in training because it uses INT4 computation while QLoRA uses NF4 computation which is not well optimized by CUDA.
> >**In the revised version, we make this point clear by adding the above contents to Section 4.2 (the paragraph starting with "the efficiency of QA-LoRA").**
>
> |Method|#Params|Time|
> |:-|:-:|:-:|
> |QLoRA ($D_\mathrm{int}=64$)|160M|40.0h|
> |QA-LoRA ($D_\mathrm{int}=64$)|89M|21.5h|
> |QA-LoRA ($D_\mathrm{int}=128)$|178M|21.8h|
>
> Table. A comparison of the number of parameters and training time.
>
> >**Q5.** Section 3.3 explains the method and the reasoning behind why the rank degenerates to 1. But the explanation is not comprehensive.
>
> >**A5.** First of all, please note that the final step of QA-LoRA is to fuse $\tilde{\mathbf{W}}$ (the quantized matrix during training) and $s\cdot\mathbf{A}\mathbf{B}$ (the adaptation weights) into a new matrix $\mathbf{W}'=\tilde{\mathbf{W}}+s\cdot\mathbf{A}\mathbf{B}$. For the efficiency of inference, we hope that $\mathbf{W}'$ is still in low-bit quantization. This means that all values in $\tilde{\mathbf{W}}$ and $\mathbf{W}'$ must come from an equidistant discrete set (see Footnote 1).
> >As an example, let a column of $\tilde{\mathbf{W}}$ contain the numerical values from the set of $\{0.3,1.5,2.7,3.9\}$, where we can use INT2 quantization ($\alpha=1.2$, $\beta=0.3$). Now, it is added by the corresponding column in $s\cdot\mathbf{A}\mathbf{B}$ (both $\mathbf{A}$ and $\mathbf{B}$ contain continuous numerical values), and we require that the summation is still INT2-quantizable. To guarantee this in a continuous optimization procedure, the only known way is to constrain the added values to be constant, which derives that all row vectors of $\mathbf{A}$ are identical, and hence $\mathrm{rank}(\mathbf{A})=1$. Actually, we can choose not to say $\mathrm{rank}(\mathbf{A})=1$ but simply say that the flexibility of adaptation is largely constrained because the row vectors of $\mathbf{A}$ must be identical.
> >We hope that we have made the explanation comprehensive. Of course, we are open to further discussions. Thanks.

---

### Public Comment · ~Sihwa_Lee1 · 2023-11-20
**Questions and appreciation regarding your research**

First of all, thank you for sharing your wonderful research results in a paper. This is a field I have been interested in, so I read your paper with great interest. I hope you don’t mind me leaving a comment during the official review period, as I have a question.

In Table 2, you compared training time with QLoRA. (Although the official source code is currently unavailable, as far as I know), when I first followed up on related research, I noticed that your source code used the AutoGPTQ library for quantized operations. However, as far as I know, QLoRA uses the bitsandbytes library, and this made me question whether the speed comparison in Table 2 is a fair comparison. This is because, to my knowledge (and based on my experience), the speed of the quantized operation kernel in AutoGPTQ is much faster than that of bitsandbytes. Therefore, I thought it was unclear whether the reported speedup is due to the advantages of QA-LoRA itself or the difference in frameworks.
Do you have any experimental results comparing speeds in exactly the same setting? Additionally, are there any experimental results regarding inference speed?

If there is anything I have misunderstood, I would appreciate it if you let me know.
Once again, I apologize for leaving a comment during the rebuttal period.

---

> ### Author Response · Authors · 2023-11-22
> **Thanks for Your Comment**
>
> Thanks for the comment. We hope that the question and our response are helpful for you and the referees to further understand the details of our paper.
>
> You are right. We used the `AutoGPTQ` library and QLoRA used the `bitsandbytes` library. This is because QLoRA was built upon the NF4 format which is *not* supported by any library except for `bitsandbytes` (created by the same authors); meanwhile, `bitsandbytes` does not support INT4 (as well as INT3 and INT2). That is why we cannot compare the two algorithms in exactly the same setting. In the inference stage, the speed of QA-LoRA is about $2\times$ of that of QLoRA (QLoRA is ran on `bitsandbytes` and QA-LoRA on `AutoGPTQ`).
>
> Regarding the advantage of training costs, we agree that QA-LoRA has a similar (slightly smaller, due to the reduced size of $\mathbf{A}$) number of operations as QLoRA. QA-LoRA's advantage in training costs mainly comes from the faster speed in executing INT4 (integer) operations compared to that in executing NF4 (floating point number) operations. This is one of the main advantages of QA-LoRA; besides, other advantages of QA-LoRA include:
> * QA-LoRA allows merging $\tilde{\mathbf{W}}$ and $s\cdot\mathbf{A}\mathbf{B}$ into one quantized matrix, making the inference easier and more efficient.
> * QA-LoRA achieved higher accuracy in the MMLU tasks, spanning over different backbones and datasets.
>
> Again, we appreciate the comments which allow us to clarify this point during the rebuttal period. Thanks.

---

### Author Response · Authors · 2023-11-22
**General Response to Area Chair and Reviewers**

We thank the area chair and the reviewers for their valuable comments. The reviewers agreed with the necessity of our research (towards making the adaptation of LLMs more efficient), the elegant implementation, and the good performance of the proposed QA-LoRA algorithm. The raised concerns mainly lie in minor issues, including explanations, technical details, and further diagnosis of the algorithm. In what follows, we address the questions point by point. We look forward to further discussions.

We also revised the paper and updated the PDF file in the OpenReview system. In the current PDF, the revisions in the main article are marked in magenta.

---

### Meta-Review · Area_Chair_WDV4 · 2023-12-04

**Metareview:**

QLoRA, in which non-quantized low-rank components are learned on top of a quantized LLM, is an effective approach for efficient LLM finetuning. However, the quantized and non-quantized component cannot naively be merged after training. This paper presents a simple approach for learning low-rank components that can be merged with the quantized component.

The main strength of this paper is the simplicity of the approach combined with some improvements (though not aways) over QLoRA baselines, as well as the fact that the resulting merged (Table 8 in the appendix). The main weakness of the paper is that while the authors claim to be motivated by inference speed-ups, this work does not actually show inference speed numbers! (Table 2 is only finetuning). This makes the paper significantly less exciting, as there are many cases where theoretical speed-ups in inference do not translate to actual speed-ups.

**Justification For Why Not Higher Score:**

Lack of actual inference speed numbers. Only marginal improvements over QLoRA.

**Justification For Why Not Lower Score:**

The method is simple, and given QLoRA's popularity, it may be of interest to many practioners.

---

### Decision · Program_Chairs · 2024-01-16

Accept (poster)